# Deploying a Novel Approach to Prepare Silver Nanoparticle *Bellamya bengalensis* Extract Conjugate Coating on Orthopedic Implant Biomaterial Discs to Prevent Potential Biofilm Formation

**DOI:** 10.3390/antibiotics12091403

**Published:** 2023-09-03

**Authors:** Shafqat Qamer, Fahrudin Che-Hamzah, Norashiqin Misni, Narcisse M. S. Joseph, Nagi A. Al-Haj, Syafinaz Amin-Nordin

**Affiliations:** 1Department of Basic Medical Sciences, College of Medicine, Prince Sattam Bin Abdulaziz University, Alkharj 11942, Saudi Arabia; 2Department of Medical Microbiology, Faculty of Medicine and Health Sciences, Universiti Putra Malaysia, Malaysia 43400, Selangor, Malaysia; gs56387@student.upm.edu.my (S.Q.); norashiqin@upm.edu.my (N.M.); narcissems@upm.edu.my (N.M.S.J.); 3Orthopedic Department, Faculty of Medicine and Health Sciences, Universiti Putra Malaysia, Malaysia 43400, Selangor, Malaysia; fahrudinch@upm.edu.my; 4Department of Medical Microbiology, Faculty of Medicine and Health Sciences, Sana’a University, Sana’a 009671, Yemen; naji2005@gmail.com

**Keywords:** silver nanoparticles, antimicrobial peptides, conjugates, biofilm formation on implants

## Abstract

This study is based on the premise of investigating antibacterial activity through a novel conjugate of silver nanoparticles (AgNPs) and antimicrobial peptides (AMPs) in line with a green synthesis approach by developing antimicrobial-coated implants to prevent bacterial resistance. The AMPs were obtained from *Bellamya Bengalensis* (BB), a freshwater snail, to prepare the nanocomposite conjugate, e.g., AgNPs@BB extract, by making use of UV-Visible spectroscopy. The antimicrobial assessment of AgNPs@BB extract conjugate was performed using the Resazurin Microtiter Assay Method (REMA), followed by the use of three biocompatible implant materials (titanium alloys, Ti 6AL-4V stainless steel 316L, and polyethylene). Finally, the coating was analyzed under confocal microscopy. The results revealed a significant reduction of biofilm formation on the surfaces of implants coated with conjugate (AgNPs@BB extract) in comparison to uncoated implants. For the MTT assay, no significant changes were recorded for the cells grown on the AgNPs/AMP++ sample in high concentrations. *Staphylococcus epidermidis,* however, showed more prominent growth on all implants in comparison to *Staphylococcus aureus*. It is evident from the results that *Staphylococcus epidermidis* is more susceptible to AgNPs@BB extract, while the minimum inhibitory concentration (MIC) value of AgNPs@BB extract conjugates and biosynthesized AgNPs was also on the higher side. This study indicates that AgNPs@BB extract carries antibacterial activity, and concludes that an excessive concentration of AgNPs@BB extract may affect the improved biocompatibility. This study recommends using robust, retentive, and antimicrobial coatings of AgNPs@BB extract for implantable biocompatible materials in accordance with the novel strategy of biomaterial applications.

## 1. Introduction

The growth in the world’s elderly population has been unprecedented in recent years, and so has the need for effective orthopedic implants with the highest success rate and minimum medical complexity. It is estimated that by 2050, the elderly population of 65 years or older will increase two-fold from 40.2 million in 2010 to 88.5 million in the United States alone. The expected number of complete joint replacement surgeries by 2030, as a result of this rise in population, will increase almost three-fold, up to 4 million implants per year [1]. A major challenge is to sustain the indwelling time of these orthopedic implants for a longer duration since about 65% of these implants are subjected to prosthetic joint infections (PJIs), which are biofilm-related infections caused by the colonization of bacteria over the implants, resulting in serious consequences, such as peri-implant infection, osteomyelitis, and septicemia, eventually resulting in the failure and removal of the implant [2].

PJI causes post-surgery complications and challenges to modern orthopedic surgery, mostly due to its high costs, risk of frequent revision surgeries, and often prolonged morbidity and death. It is a kind of biofilm-mediated infection that penetrates the implant, especially during the first two years after the arthroplasty surgery [3]. It is theoretically justified because the soft tissue healing process is still taking place, and there is a risk of postoperative inflammation [4]. Another cause is the small number of bacteria colonizing within the joint, resulting in a diminished ability of pathogens to evoke a strong immune response. As a result, biofilm-mediated PJIs often necessitate surgical revision and prosthesis replacement, which might result in a higher risk of infection recurrence and severe morbidity [5,6]. Patients with early PJIs also suffer from perioperative contamination, wound healing difficulties, or postoperative hematoma formation. Additionally, PJI is also known to be the most common cause of total knee arthroplasty (TKA) failure, as well as an increase in the number of total hip arthroplasties (THA), due to lack of protective soft tissue coverage, and increased stress due to less mobility of joint and soft tissue. The type of bioimplants used to treat PJI or inflammation caused by biofilms [7] are made of biocompatible materials, including metals, polymers, ceramics, composites, and other natural materials [8]

Biofilm formation on implants is often the result of the presence of *Staphylococci species* at the surgical site, which accelerates the process of biofilm adhesion over the prosthesis. Two strains, namely *Staphylococci aureus* and *Staphylococci epidermidis,* cause implant triggering and implant loosening. The *Staphylococcus aureus* strain particularly records a high rate of antibiotic resistance and plays a crucial role in implant infection; furthermore, there is a tremendous increase in the antibacterial resistance of bacterial species [9]. Eventually, a colony of bacterial cells is formed, which builds a self-produced matrix comprising proteins, polysaccharides, and extracellular DNA [10,11]. Studies reveal that strains of *Staphylococci epidermidis* and *Staphylococcus aureus* cause up to 40% of orthopedic postsurgical and implant-related infections and are found to be resistant even to gentamicin.

It is important to check the failure rate of orthopedic implants due to PJI or other similar infections and reduce the number of revision surgeries. A wide spectrum of research is now being carried out to modify the surface of implants, prevent bacterial adhesiveness, and delay the formation of biofilms on implants. Studies have recommended antimicrobial strategies based on one of two methods: either to initiate some consistent antimicrobial mechanism by modifying the implant surfaces, directly or indirectly, or to devise and implement antimicrobial action with the help of biocides embedded over the implant surfaces [12,13].

Surfaces commonly used for orthopedic implants include titanium alloys, (Ti 6AL-4V) stainless steel (316L SS), cobalt-based (Co-Cr) alloys, various polymeric biomaterials (ceramics, hydroxyapatite or polyethylene), and polymethyl methacrylate (PMMA) or bone cement, all of which reflect structures that are likely to be colonized, resulting in the formation of bacterial biofilm [14]. There are a few other metals that have been used for bioimplant applications, such as nickel titanium (NiTi) and tantalum (Ta). These metals are used for bioimplant applications due to their shape and memory properties [15]. In such cases where these metals are used, there is a need to implement tissue debridement and surgical removal processes. However, it is observed that the infection rate of follow-up surgeries in all such cases often rises to 40% [16].

Research on antibacterial AgNPs@BB extract coating on titanium, stainless steel, and polyethylene implants so far has not been subjected to clinical applications. Studies conducted so far have only highlighted AgNPs enhancing the osteogenic capacities of titanium-assisted implants [17]. These studies, however, did not analyze the antibacterial properties of these implants, a research gap that was to be evaluated in the light of bone infection models [18]. There was also a need to give more attention to adopting new alloys with superior corrosion resistance properties.

The current study was, therefore, premised on developing a new antimicrobial agent for preventing biofilm-related infections in orthopedic implants, which could not only assist in overcoming the drug resistance among pathogens but also prevent the biofilm formation on biomaterial surfaces of metallic implants. This study aimed to examine the most common reasons behind metallic implant failures and discover novel approaches to mitigate corrosion in metallic implants.

This study also aimed to identify material that could inhibit infection on implants post-treatment or post-procedure. Several novel approaches are being attempted to revolutionize orthopedic implant surfaces. For example, the use of nanoparticles (NPs) on orthopedic implants to prevent infection is gaining a lot of attention because of their quantum properties, their potentiality to absorb and sustain other compounds, and their size, which is larger than any other particle. In addition, NPs are largely involved in activities like antibacterial, anticancer, anti-inflammatory, antifungal, and venom neutralization [19]. In addition, the nanoparticles are recognized as highly biocompatible materials as they are corrosion-resistant in biological environments. By employing NPs, it is also possible to reduce antibiotic resistance, enhance antibacterial activity [20], rejuvenate older existing antibiotics, and replace them with antimicrobial peptides, provided they are aptly conjugated.

Another novel approach refers to a combination of biological agents with AgNPs that could enhance antimicrobial properties and eliminate any possibility of biofilm-related infections [21]. Once such novel class of biological compounds is antimicrobial peptides (AMPs), which are currently being utilized and have emerged as an alternative to contemporary antibiotics due to their amphipathic properties and/or hydrophobic nature. AMPs are a kind of positively charged small peptide with broad antimicrobial properties. There are studies related to the structure activity of these peptides that reveal two major requirements for antimicrobial properties: (1) the presence of a positive charge and (2) an activated amphipathic conformation. The action mechanism of these AMPs is being actively researched, and the available information is growing. The majority of these previous studies have concentrated on the relationship of positively charged peptides with model membrane processes [22,23]. Owing to being small-sized peptides containing 20–60 amino acid residues, AMPs do not damage the cell membrane but have a positive charge, with 30% hydrophobic residues and a potential to inhibit DNA replication to cross the bacterial cytoplasm [24]. There is extensive use of innovative AgNPs bone cement-coated devices, such as external fixation pins, proximal femur, or tibia megaprostheses, which exhibit an infection inhibition trend in the field of orthopedics.

In the current study, we have developed special orthopedic implant biomaterial discs. A novel approach of coating was adopted to make the developed implants potential against biofilm formation. In order to develop nanostructured functional coatings having anti-adherent properties and the potential to avoid bacterial cell adhesion, our research is premised on the combined use of *Bellamya bengalensis* (BB) extract and AgNPs. To the best of our knowledge, there are only a few studies available on the surface coating of implants with silver nanoparticles conjugated with BB (AgNPs@BB) [25,26]. In addition, there is a dearth of studies on the biosynthesized AgNPs@BB extract coating method used in the current study, the method of a dipping coating technique on the surfaces of titanium alloys (Ti6AL-4V), stainless steel (316L), and polyethylene. The coating method recommended in this study is very simple, economical, and viable in the distribution of AgNPs@BB extract for avoiding biofilms and bacterial activity. To collect further evidence of this coating technique, the surface morphology of coatings and their effects on biofilms and antibacterial activity of the coating were investigated with a focus on *Staphylococcus aureus* and *Staphylococci epidermidis* infections, which was highly cumbersome, particularly in the case of orthopedics infections.

There is also insufficient evidence to support the use of silver-coated implants or AgNPs@BB extracts as a therapeutic alternative or a substitute for antibiotics that could check the biofilm formation on implants. A need was therefore felt to investigate whether, despite the antimicrobial properties, a conjugation such as AgNPs@BB extracts would create a synergistic effect and lead to the elimination of bacterial infections on orthopedic implants. To investigate this phenomenon and find evidence of these premises, the following research question was framed for this study: how can biosynthesized AgNPs@BB extract, prevent, or eliminate bacterial infections related to biofilm formation on orthopedic implants made of titanium (Ti), stainless steel (SS), and polyethylene (PE)?

Such conjugates of AgNPs with AMP structures have already been proposed in past studies, focusing on procuring such biomaterials or implant coatings that have high biocompatibility and antibacterial properties, and that have recommended developing a conjugate of BB extract with AgNPs against methicillin-resistant species, such as *Staphylococcus aureus* and *Escherichia coli* (*E. coli*) [27]. A few studies have recommended the use of silver hydroxyapatite (AgHAP) for coating the implant surfaces, preventing postoperative infection, and increasing bone fusion capacity [28] or employing AgHAP to reinforce nanocomposites of poly(methyl methacrylate)-poly(ε-caprolactone) as hybrid orthopedic materials [29]. Hence, the current study aimed to advance these research initiatives in the context of implants and examine whether such a conjugation of metal particles with antimicrobial peptides could prove antimicrobial safety and biocompatibility (osteoconductivity). In particular, this study proposed a conjugation of BB extract with AgNPs (AgNPs@BB extracts) to initiate an increase in the localization of peptides at the infection site and facilitate the targeting of the peptides more efficiently.

## 2. Materials and Methods

### 2.1. Materials

*Bellamya bengalensis* was procured as a potent source of protein from freshwater snails. These were accessible and procured from the local market of Riyadh, Saudi Arabia. Silver nitrate (AgNO_3_) was purchased from Merck, Darmstadt, Germany. The host department of the researcher, Department of Microbiology, Prince Sattam bin Abdulaziz University, Alkharj, Saudi Arabia, provided the bacterial strains of *Staphylococcus aureus* (ATCC^®^ 25923) and *Staphylococcus epidermidis* (ATCC^®^ 51625).

*Hemigraphis colorata* leaves were collected from the Royal Commission, Riyadh City (Riyadh, Saudi Arabia), in February 2022. These leaves were identified and verified at Prince Sattam bin Abdulaziz University (Alkharj, Saudi Arabia). *Hemigraphis colorata* plant grows in abundance in areas of Riyadh, Saudi Arabia, as natives use it for the treatment of wounds. The plant is identified by its color, with the upper region being dark green and the lower region green or purple.

The implant disks made of titanium, stainless steel, and polyethylene were obtained from Vishal Surgitech, Gujrat, India. The ingredients of these disks included titanium grade–5 (Ti6AL-4V), stainless steel (315L), polyethylene (PE), and orthopedic implants (ISO 5832-3) (lot number L0011011, manufacturing date May 2022, expiry date not applicable).

### 2.2. Methods

#### 2.2.1. Preparation of Leaf Extract

First and foremost, dust particles were removed from the fresh and mature *Hemigraphis colorata* leaves. A sample of 20 g of fresh leaves was carefully washed in de-ionized fluids and air-dried at room temperature. These leaves were then coarsely put into an electric grinder to obtain small fibrous flakes, which were further refined to micro/nano size through a high-energy ball milling process, which enabled dry pulverization. Zirconium balls (10 mm) and a sintered corundum container (80 mL) were used to dry up the milling. This dried-up powder was stored at room temperature in plastic bags until its use. Prior to its use, the powder was washed again and agitated for 20 min at 60 °C. After boiling, the leaf extracts were cooled to room temperature and filtered, yielding 75 mL of leaf broth that was kept at 4 °C. The flow process for the extraction of leaf extract is shown in Figure 1.

#### 2.2.2. Biosynthesis of Silver Nanoparticles

The biosynthesis of the AgNPs takes place when the extracts of *Hemigraphis colorata* are conjugated with silver nitrate. Hence, for the next step, leaf extracts of *Hemigraphis colorata* were mixed with 1 mg/mL in water. This extract was partially soluble in water. Next, 5 mL leaf broth was mixed with 45 mL of AgNO_3_ aqueous solution and left to react at room temperature. This resulted in a color change in the mixture, from colorless to dark brown, which was an indication that AgNPs were being formed. At this stage, centrifugation at 5000× *g* for 20 min was conducted to collect the AgNP precipitates. The pellets were collected and dried for 1 h in a hot air oven at 50 °C. The powders 1 mg/mL of each NP were dried and stored in microcentrifuge tubes for later analysis. These results are consistent with prior studies, which also reported that *Hemigraphis colorata* extracts changed their color when added to AgNO_3_ solution, which was attributed to the reduction mechanism of Ag+ to Ag^0^ [28,29]. These results also validate the enhancement of bio-compatibility of AgNPs along with their secondary metabolites embedded in plant extracts [30].

#### 2.2.3. Extraction of Antimicrobial Peptides from *Bellamya bengalensis*

*Bellamya bengalensis*, obtained from freshwater snails, was available in abundance locally. To prepare the extract, snails were rinsed in distilled water until their shells could be cracked. After cracking their shells, their soft meat was stored at 20 °C, homogenized in a blender with 10% ice-cold acetic acid solution (1:3 *w*/*v*) (1 gm snail tissue and 3 mL of 10% acetic acid). This homogenized solution was centrifuged at 4 °C for 30 min at 5000× *g*, enabling the precipitation of soluble proteins for 1 h at 4 °C with ammonium sulfate (NH_4_)^2^SO_4_ at 100% saturation and continual stirring. The precipitation was suspended in 10 mm Tris-HCl buffer, pH 7.0, after centrifugation at 5000× *g* for 30 min at 4 °C. Lastly, the peptides were isolated and kept at 4 °C and dialysis buffer was prepared (50 mm Tris-Cl at pH6.8 and 100 mm NaCl). The dialysis bag was filled with the precipitates and placed on the dialysis buffer. Stirring at 4 °C continued overnight, and protein samples were again collected by centrifugation at 13,000× *g* for 20 min at 4 °C. The supernatant was collected and used for further analysis.

#### 2.2.4. Purification of Peptides Using Ion Exchange Chromatography

The chromatography column enables the preparation of DEAE cellulose (diethylaminoethyl cellulose, MB110-5G) with a 4 cm thick bed. To perform our test, ethanol was used to wash before each use, using a 20 mm Tris-Hcl, and prepare a 100 mL volume, attuning pH to 8.5 after using NaOH. For the purpose of preparing the bed, counter ions (salt gradient) of 40 mL of 25 mm NaCl in 50 mm tris (pH-7.2) and 0.3% of DEAE were applied to the column. The elutes, using sodium phosphate citrate buffer (pH-7.2), were run for 4 h to collect 2 mL and collected in a beaker. Post dialysis, the crude AMPs were poured into the column without disturbing the bed and left for 20 min to settle. Next, the first eluting buffer, i.e., sodium phosphate citrate buffer (pH-7.2), was loaded into the column and used to elute the sample. The column was finally allowed to settle for 15–20 min when the elute was stored at freezing conditions.

#### 2.2.5. Preparation of AgNPs@BB Extract Conjugates

The dialyzed peptides were lyophilized, and the water contents were removed. AgNPs@BB extracts were performed in a 1:1 combination. For a 1:1 ratio, 100 mg of antimicrobial peptides (AMP) were dispersed in 1 mL of sterile distilled water (pH 7) and added drop wise to 100 mg AgNPs. The solution was incubated at room temperature for 30 min. The conjugate formed spontaneously, as verified by UV-Vis spectroscopy and scanning electron microscopy. Since there was no need for any specific incubation time, it was very convenient to remove the excess peptide by allowing the solution to flow through a 10 kDa membrane. It has been recommended that a buffer exchange be carried out with phosphate-buffered saline (pH 7.4) [31]. Figure 2 shows the schematic for the preparation of AgNPs, BB extract, and AgNPs@BB extract composites.

#### 2.2.6. Preparation of Biomaterial Discs

In accordance with the industry standards, laboratory procedures, and recommendations of the manufacturer, the biomaterial rods of titanium (Ti) grade-5 Ti6AL-4V, stainless steel (SS) 316L, and polyethylene (PE), (10 H) of length 167 mm, were cut into discs (thickness of 2–3 mm) with the Buehler IsoMet 1000 machine (Buehler, Lake Bluff, IL, USA). Each specimen of Ti6AL-4V, 316L, and PE was 4.5 mm broad in diameter and 2 mm in thickness, following the dimensions used in clinical practices. All these rods were obtained from Vishal Surgitech, Gujrat, India.

#### 2.2.7. Coating of AgNPs@BB Extract Composites on the Orthopedic Implants

Complying with the recommended practice, AgNPs@BB extract composites were coated on orthopedic implants by making slight modifications of methods as described in a previous study [32]. For the first step, implants were cut into small pieces and sterilized by autoclaving at 121 °C. These implants were then dipped into a coating solution of a mixture of AgNPs@BB extract conjugates (50 mg/mL for 1× and 100 mg/mL for 2×) of molten polyethylene glycol while being subjected to stirring conditions. Subsequently, heating of the mixture at the range of 60–70 °C in a water bath was conducted to obtain a homogeneous slurry. When this step was complete, implants were immersed in the slurry for 15 min and were washed in sterile distilled water. At this stage, any loosely attached coating material was removed from the surface of the coated segments. In the end, the pieces were left to air dry overnight at room temperature on a sterile surface under an aseptic condition in the air of laminar flow [33]. Figure 3 presents the scheme for the coating of AgNPs@BB extract composites on implants.

### 2.3. Characterizations

#### 2.3.1. Surface Characterization

UV-Visible spectroscopy (UV-Vis) (Shimadzu, UV-1280, Kyoto, Japan) in the range of 200–800 nm was used to investigate the physicochemical characterization of the nanocomposites. In addition, Zetasizer at 25 °C was used to determine the hydrodynamic size range and polydispersity index (PDI) of dispersed AgNPs@BB conjugates, with a scattering angle of 90° (Malvern Instruments Ltd., Malvern, UK). A field emission scanning electron microscope (FESEM) (CARL ZEISS, Jena, Germany) was used to study the surface morphology using an accelerating voltage of 5 KV. An emission current of 12 mA (up to 66,000×) was used to analyze the shape and size of AgNPs. TEM tests were performed at 200 KV, which helped to determine the morphology, size, and shape of the AgNPs via HITACHI H-800. To analyze the functional group present in the AgNPs and AgNPs@BB conjugates, FTIR tests were also conducted in KBr medium using a Thermo Nicolet Avatar 370 model FTIR spectrometer with a resolution of 4 cm^−1^ in the range of 400–4000 cm^−1^.

#### 2.3.2. Antimicrobial Assay of Silver/BB Composite Using Resazurin Microtiter Assay Method (REMA)

In an in vitro study, the antimicrobial activity is determined by the use of a modified broth prepared through a microdilution method. This method incorporates resazurin as an indicator of cell growth in 96-well microtiter plates. The minimum inhibitory concentration (MIC) biosynthesized AgNPs@BB extracts were conducted as per the CLSI M07-M-11 guidelines [34]. The bacterial inoculums were then adjusted according to the concentration of 10^6^ CFU/mL. The susceptibility of Gram-positive *Staphylococcus aureus* (ATCC^®^ 25923) and *Staphylococcus epidermidis* (ATCC^®^ 51625) bacteria strains was investigated. To perform the MIC test, 100 µL of AgNPs, BB extract, and AgNPs@BB extract composite concentration from 1.9 to 500 mg/mL in three microtiter plates was used as a control (50 μg/mL) element, ensuring that bacterial inoculums of 100 µL MHB start from column 1 to column 12. Each well of the microtiter plate was mixed with 30 µL of the resazurin solution of an in vitro toxicology assay kit (code: 263-718-5) according to the manufacturer’s recommendation (Sigma–Aldrich, St. Louis, MO, USA) and incubated at 37 °C for 24 h. Color changes were observed: blue/purple indicated no bacterial growth, while ink/colorless indicated bacterial growth. Tests were performed in triplicates, and graphs were plotted [35].

#### 2.3.3. Bacterial Biofilm Formation on Biomaterial Discs

A 6-well microtiter plate was used while the wells were inoculated with 0.5% McFarland standard solution of the bacterial strains used in this experiment (*Staphylococcus aureus* (ATCC^®^ 25923) and *Staphylococcus epidermidis* (ATCC^®^ 51625)), by scaling each inoculum to the value of 10^5^ CFU/mL.

At this stage, dry-heat sterilized titanium, stainless steel, and polyethylene discs were inserted separately into 500 mL of nutrient broth (NB), supplemented with 0.25% glucose in conical flasks. These discs were inoculated with *Staphylococcus aureus* and *Staphylococcus epidermidis* to a final concentration of 10^5^ CFU/mL and cultured for 48 h at 37 °C in a shaker incubator at 160 g. The discs were aseptically removed from the microtiter plate, and washed gently three times in 10 mL × phosphate-buffered saline (PBS) buffer (7.4 pH) (Gibco™, Life Technologies, Bleiswijk, The Netherlands code; 12579099) to remove non-adherent cells, and then transferred to 50 mL PBS.

#### 2.3.4. Confocal Analysis of Biofilm Inhibition

In the next stage of the experiment, biofilms on the surfaces of the titanium, stainless steel, and polyethylene discs were examined for bacterial viability using fluorescence microscopy. These discs were treated with SYTO-9 from the Live/Dead^®^ Backlight TM bacterial viability kit (code; S-34854) (Life Technologies, Carlsbad, CA, USA) and kept at room temperature in the dark for 15 min, as per the manufacturer’s instructions. The advantage of SYTO-9 (green stain) is that it can penetrate any live or dead Gram-positive and Gram-negative bacteria, i.e., bacteria with intact and disrupted membranes. These discs were sterilized and uncoated (controlled), combined with organisms, but without the use of AgNPs@BB extract. In order to examine under a confocal laser scanning microscope (CLSM, Olympus, FV1200, Shinjuku-ku, Tokyo, Japan), the stained biofilms were gently washed with PBS.

This step ensured the prevention of the biofilm-associated substrates from drying during imaging. To achieve this, the samples were soaked in a clean 0.85% NaCl solution during imaging. After imaging, the surfaces with biofilms were transferred to 0.85% NaCl solution and pre-warmed at 40 °C for 10 min to trigger shape recovery. After the shape change at 40 °C for 10 min, the substrates were gently washed three times again with a clean 0.85% NaCl solution, and digital images were captured. The excitation and emission wavelengths used for detecting SYTO-9 were 488 and 525 nm, respectively, with objective lenses of ×2.5 and ×10. The stained biofilms were analyzed with Image J software (version 1.49 g). Three-dimensional stack images of the biofilms were analyzed to quantify bio-volumes of biofilms. At least three biological replicates were tested for each condition.

#### 2.3.5. Inhibition of Biofilm Formation (Microtiter Plate Assay)

This stage commenced with the crystal violet staining of the colorimetric assay of biofilms [36]. The microtiter plate technique was used to assess the inhibitory activity of AgNPs@BB extracts conjugates against biofilm formation. To begin with, *Staphylococcus aureus* and *Staphylococcus epidermidis* were cultivated in tryptic soy broth (TSB). A total of 100 μL of inoculum with 0.5% McFarland turbidity was supplemented with 10 μL of AgNPs@BB extract conjugate at concentrations of 0.5 μg/mL,1 μg/mL, and 2 μg/mL. This mixture was then inoculated into a 96-microtiter plate at 37 °C for 24 h to allow biofilm formation. The plate was then removed by adding 100 μL 0.9% (*w*/*v*) NaCl and washed three times to eliminate the free-floating bacteria. It was understood that biofilms had already taken the form of adherent cells on the surface of the plate. Hence, it was necessary to fix them with 200 μL of 95% ethanol and stain them with 0.1% crystal violet for 15 min. The wells of the microtiter plates were finally washed 3 times with 300 μL of 0.9% (*w*/*v*) NaCl to remove the dye and left for air drying.

Subsequently, 160 μL of 33% glacial acetic acid was transferred to each well to suspend the crystal violet from stained biofilm, and the optical density was read at 570 nm (spectrophotometer Biotek, Peabody, MA 01960, USA).

#### 2.3.6. Cytotoxicity Analysis of AgNPs@BB Extract

For cytotoxicity of AgNPs@BB extracts, mouse fibroblastic cells (L929) were cultured, following the supplier guidelines. The cytotoxicity of the synthesized AgNPs@BB composite was assessed by MTT (3-[4,5-dimethylthiazole-2-yl]-2, 5-diphenyl tetrazolium bromide) dye conversion assay [37,38]. Next, L929 cells at a density of 1 × 10^4^ per well were cultured in 100 μL of cell culture medium (DMEM: Dulbecco’s Modified Eagle Medium) supplemented with 10% fetal bovine serum in a 96-well cell culture plate. After 24 h, cultured cells were treated with a series of concentrations (5, 25, 50, 75, and 100 μg/mL) of filter-sterilized AgNPs@BB extracts composite in 100 μL/well (culture medium: DMEM without serum) and incubated further for 24 h. This was followed by the removal of the medium and treatment with MTT dye at a final concentration of (0.5 mg/mL) and further incubated for 4 h. Finally, 100 μL of dimethyl sulfoxide (DMSO) was added to each well to dissolve blue formazan precipitate, and absorbance was measured at 570 nm using a microplate reader (Bio-Rad Model 680; Bio-Rad, Gurugram, India). The cytotoxicity was calculated by comparison of ODs with the positive control.

For the enumeration of dead cells, acridine orange (AO) staining was applied to both treated and untreated live and dead cells (L929). In this study, 5 μg/mL and 100 μg/mL of AgNPs@BB extracts were taken as minimum and maximum concentrations, respectively, for the toxicity study. The documentation of cell viability was carried out under an Inverted Phase Contrast Microscope (Eclipse TS100, Nikon, Tokyo, Japan, Am Rosengarten 5, 14621) equipped with a fluorescence unit and digital camera (Coolpix 5400, Nikon).

### 2.4. Statistical Analysis

Antimicrobial activity and MIC analysis were performed in triplicate, and the comparative results were represented as mean ± standard deviation with *p* < 0.05 * as a significant difference. One-way ANOVA was used for the comparison of quantitative variables by using SPSS version 35 software, while the values of One-way ANOVA were measured as F = 145.657, df = 2, SD 0.941, and Mean 3.47.

### 2.5. SEM Analysis of Uncoated and Coated Implants

SEM analysis was performed on all implant samples uncoated (titanium, polyethylene, stainless steel), coated with 1× (50 mg/mL) concentration (titanium, polyethylene, stainless steel), and coated with 2× (100 mg/mL) concentration (titanium, polyethylene, stainless steel). After the coatings with concentrations (50 mg/mL and 100 mg/mL), all the samples were rinsed in ethanol solution. The purpose of rinsing was to stabilize the coating of AgNPs@BB extract conjugates on implant surfaces.

## 3. Results and Discussion

### 3.1. UV Visible Spectroscopy

UV-Vis is a process to analyze the extinction (scatter + absorption) of the passage of light through a substance. It is a great tool to recognize, characterize, and investigate nano-materials that have optical properties that are sensitive to size, shape, concentration, aggregation state, and refractive index. The bio-reduction of Ag+ from AgNO_3_ into Ag0 by the secondary metabolite from *Hemigraphis colorata* leaf extract is the fundamental step in the synthesis of AgNPs [39]. The UV-Visible spectrum of the AgNPs that were biosynthesized is shown in Figure 4a as evidence of bio-reduction. Some of the peaks in the spectrum of leaf extract fall between 200 nm and 400 nm, indicating that the secondary metabolites of leaf extracts contain phenolic compounds [40]. These peaks associated with leaf compounds vanish when Ag+ is reduced, leaving only a single peak at 400 nm, which is the surface plasmon resonance spectrum of AgNPs [41].

The use of leaf extract in the synthesis of AgNPs yielded a similar spectrum as it could produce biochemically in antimicrobial proteins experimented for Gram-positive bacteria [42], or when there is an improved efficacy against multidrug-resistant bacteria [43]. Due to their strong surface plasmon resonance (SPR), AgNPs are well-known to exhibit UV-Vis absorption in the 400 nm–500 nm range. The conjugates of AgNPs@BB extract exhibited specific absorption at 395 nm, while AgNPs extracted exhibited specific absorption at 403 nm (Figure 4b). The decrease in intensity of AgNPs@BB extract was due to the strong binding of peptides with AgNPs, in agreement with the previous study [44].

### 3.2. Physicochemical Characterization of AgNPs

The well-known technique for imaging solid materials at atomic resolution is TEM. The technique was employed to visualize the size and shape of silver nanoparticles and conjugates of AgNPs@BB extract. The TEM image in Figure 5a,b represents a heterogeneous form of the nanoparticles, with the particle sizes ranging from 36 to 54 nm. It was also observed that most of the AgNPs were spherical.

In the analysis of AgNPs@BB conjugates, the formation of conjugates was clearly observed. They were found to be irregular in shape and unevenly distributed. The size of the AgNPs@BB extracts was in the range of 67.43 nm to 90 nm. The AgNPs@BB extracts were found to be increasing in shape, bigger than the AgNPs. The bio-organic layer of peptides was also observed. Transmission of electron microscopy image also led to the aggregation of the conjugates and interaction of hydrophobic organic peptide oligomers. These hydrophilic interactions mediated the arrangement of AgNPs@BB extracts into spherical assemblies to increase their size and serve as templates. Therefore, conjugates of AgNPs with peptide oligomers not only bring about the formation of aggregates, but also have a role in controlling their morphology [45].

Moreover, the particle size of AgNPs and the conjugates of AgNPs@BB extract were further justified by comparing the results with the average sizes obtained from the particle size distribution. The particle size distribution implies a particle size mean of around 51 nm for AgNPs, 77 nm against AgNPs@BB extract and PDI index, AgNPs (0.251), and AgNPs@BB extract (0.62), respectively. The value of the PDI index in Zetasizer software lies between 0 and 1. Hence, the PDI must be less than 1 for samples to be suitable for DLS analysis. The decrease or increase in PDI might be due to the reduction or enhancement in surface tension of AgNPs after being functionalized with AMP. Therefore, enhanced PDI of AgNPs with conjugate (BB extract) also assures the loading of AMP on the AgNPs surface [46].

TEM analysis was performed by the synthesized AgNPs of the heterogeneous size of each particle ranging between 36 and 54 nm. Moreover, the size of AgNPs was further justified by comparing the results with the average sizes obtained from the particle size distribution presented in Figure 5c,d. The particle size distribution implied a particle size mean of around 51 nm for AgNPs, and their zeta potential was found to be −29.1; this indicated that the biosynthesized NPs were highly stable. Though NPs were evenly distributed, their size was slightly varied between AgNPs@BB extracts, suggesting the presence of distinct biomolecules in each extract and allowing for the correlation of nanoparticle size with the results obtained from TEM results [47].

### 3.3. FTIR Analysis

FTIR spectrum of green synthesized silver nanoparticles (AgNPs) provided different absorption peaks, which assigned the different functional groups of phytochemical compounds. In this process, the AgNPs were characterized by FTIR analysis according to their peaks. Three peaks were observed at 3310 cm^−1^, 1637 cm^−1,^ and 671 cm^−1^ (Figure 6a). The intense broadband at 3310 cm^−1^ indicated the presence of N-H and O-H stretching modes of the phenolic compound [48]. It revealed the presence of phenolic acids or flavonoids. The absorption band was noticed at 1637 cm^−1,^ arising due to carbonyl (CO) groups [49]. The primary phytochemical screening also showed that the leaf extract of *Hemigraphis colorata* was rich in phenol, carbohydrates, tannins, proteins, carboxylic acid, and flavonoids. These components were observed causing N-H stretch of primary and secondary amines and amides, C–H stretch of methyl groups, H-C=O stretch of aldehydes, C=-N stretch of nitriles, C=-C stretch of alkynes, and C=-O stretch of carbonyl groups of flavonoids and tannins, respectively [48,50]. The narrow bands at 671 cm^−1^ and 602 cm^−1^ were attributed to glycoside and substituted phenolic compounds present in the AgNPs. The intense and broad peak at 432 cm^−1^ corresponded to the Ag metal [51,52]. Therefore, the current findings of FTIR showed that the phenolic proteins acted as the capping agent for AgNPs, and played a significant role in increasing the stability of NPs synthesized.

Figure 6a,b shows major peaks noted at 3286.70 and 1635.64 cm^−1^ in the spectra of the AgNPs@BB extracts, indicating the presence of polyphenolic-OH groups and N-H stretching of amines, as detected values were approximately 3300 cm^−1^ [51,52]. Other intense peaks noted at 679 and 602 cm^−1^ for AgNPs@BB extracts indicate amide 1 and carbonyl (C=O) stretching of proteins [53]. There are varied biomolecules, such as proteins and polyphenolics, which are used in all tested solutions, where they demonstrate the role of a capping and stabilizing agent in the AgNPs fabrication process. There is also a vibration of C=C or the asymmetric stretching of N-O, which originates from AMP and some narrow beams at 432 cm^−1^ and 486 cm^−1^, which undergo a shift to 416 cm^−1^, 468 cm^−1^ due to the formation of AgNPs@BB extracts conjugates [54].

Moreover, the decrease in intensities in peaks in AgNPs@BB extracts spectra at 1635.64 cm^−1^ reflected the interactions between the atoms of Ag, O, and N. This peak is often attributed to the imine group (C=N). The absorption bands located at approximately 3286.70 cm^−1^ are found in the spectra of all analyzed samples, which can be associated with the -C=O group. The appearance in the spectrum of AgNPs@ indicates the presence of peaks at 678 cm−^1^, 601 cm^−1^, 563 cm^−1,^ and 489 cm^−1^, suggesting that the conjugate was successfully incorporated into films [55]. The above results are further summarized in Table 1.

### 3.4. Minimum Inhibitory Concentration

Minimum inhibitory concentration (MIC) was determined by the Resazurin Microtiter Assay Method (REMA), which is a micro-broth dilution technique against Gram-positive *Staphylococcus aureus* and *Staphylococcus epidermidis* bacteria. The proposed mechanism of antibacterial activity of silver nanoparticles varies from bacterium to bacterium and from peptide to peptide [56] or is revealed as a bacterial membrane with cell wall disruption or bacteria invasion, showing oxidative stress [57]. Such antibacterial activity often results in an electrostatic interaction, causing positively charged AgNPs to attach to cell membranes [58]. However, in the current study, since it was applied for antibacterial activity, the MIC values (Table 2) were to be determined. Table 2 exhibits the values wherein AgNPs@BB extracts composite exhibit an effective antibacterial activity against *Staphylococcus aureus* and *Staphylococcus epidermidis* (MIC of 3.9 ± 0.04 μg/mL and 3.9 ± 0.01 μg/mL, respectively), which corroborates that the composite has the highest antibacterial effectiveness than AgNPs and BB extract. It was also observed from the obtained MIC values that the MIC of AgNPs was higher than the MIC of BB extract. For the silver nanoparticles, the MIC against *Staphylococcus aureus* was found to be 7.8 ±0.67 μg/mL, while for *Staphylococcus epidermidis* it was 12.8 ± 0.67 μg/mL. The MIC values for the BB extract against *Staphylococcus aureus* and *Staphylococcus epidermidis* were determined to be 31.2 ± 0.01 μg/mL and 15.6 ± 4.3 μg/mL, respectively. Statistical analysis revealed a significant difference between the two groups, with a *p*-value of less than 0.05 *.

AgNPs@BB extract conjugate exhibited lower MIC values and were more effective in inhibiting the growth of *Staphylococcus aureus* and *Staphylococcus epidermidis* compared to the AgNPs and BB extract, as exhibited in Table 2. The MIC values of BB extracts and AgNPs suggested that the antibacterial activity of AgNPs was higher than the antibacterial action of AgNPs@BB extract conjugate. This could be explained well by considering the major factors that affect the action mechanism of silver nanoparticles towards bacteria, such as their unique size, topology, shape, morphologies, and surface charge. Furthermore, the highest antibacterial effectiveness exhibited by the AgNPs@BB extract composite could be ascribed to the synergistic effect of the strong antibacterial action of both AgNPs and AMPs.

### 3.5. SEM Micrographs Analysis

In comparison with all uncoated sample discs, the surface of the uncoated titanium implant (shown in Figure 7a) was corrugated and found uneven as compared to uncoated polyethylene and stainless steel, while the uncoated surfaces of polyethylene and stainless steel were quite smooth, as shown in Figure 7d,g.

In the same way, in comparison with all coated implant samples, there was more dense coating in the case of titanium (Ti) (Figure 7b,c) compared to coated polyethylene (Figure 7e,f) and stainless steel (Figure 7h,i). Actually, titanium (Ti) displayed an evident fissure due to the existence of micro-fractures, which tended to attain denser and thicker coatings. Secondly, a dense coating was found for polyethylene surfaces compared to stainless steel. In fact, steel is more smooth, polished, and plainer compared to synthetic polymer (polyethylene), due to its roughness, corrugation, and microfractures. These features provide more active sites and surface area for the deposition of coatings [59]. Scanning electron microscopy also revealed the homogeneous deposition of AgNPs@BB extract conjugates on all surfaces. Moreover, with an increase in concentration from 1× to 2×, the deposition of AgNPs@BB extract conjugates was found to be more uniform and denser. This further indicated the higher tendency of formation of a coating network when concentration was increased.

### 3.6. Confocal Analysis

The confocal laser scanning microscopy (CLSM) analysis of all implant samples uncoated (titanium, polyethylene, stainless steel), coated with 1× concentration of AgNPs@BB extract conjugates (titanium, polyethylene, stainless steel), and coated with 2× concentration of AgNPs@BB extract conjugates (titanium, polyethylene, stainless steel) was conducted. The analysis was performed to evaluate the biofilm inhibition potential of the AgNPs@BB extract conjugates. SYTO-9 dye was used for the staining of biofilms. It was observed that SYTO-9 dye especially stained the living cells, representing the biofilm formation on the surface of the implants. A number of medical studies have shown prominent growth of *Staphylococcus epidermidis* on implants as compared to *Staphylococcus aureus* [60].

The 3D images of Figure 8a–c show the thickness of *Staphylococcus epidermidis* growth (90 µm, 60 µm, and 40 µm). There was higher growth of *Staphylococcus epidermidis* on the titanium implant compared to the other two implants (polyethylene and stainless steel), because the surface of all coated sample discs (as shown previously in Figure 7a) were corrugated and uneven compared to uncoated polyethylene and stainless steel. While the uncoated surfaces of polyethylene and stainless steel were quite smooth, as shown previously in Figure 7d,g, the surface area of titanium implants was more so and provided adequate space for the growth of microbes compared to plane surfaces. Afterward, the growth of *Staphylococcus epidermidis* was observed more in polyethylene compared to stainless steel. Again, it happened because polyethylene was a synthetic polymer (less smooth surface, flexible, and having micropits) compared to steel (more smooth, hard, and slippery against microbes) [60]. The overall green color further demonstrates biofilm formation on all uncoated implants. In addition, the biofilm formation also depends upon the surface properties of the implanted material.

Figure 9a–f shows the 2D and 3D confocal microscopic analysis of coated implants with AgNPs@BB extract concentrations (50 mg/mL). Figure 9a–c is the 2D and 3D confocal images of coated implants with 1× concentrations (titanium, polyethylene, and stainless steel) against the biofilm formation of *Staphylococcus epidermidis* , and Figure 9d–f) are the 2D and 3D confocal images of coated implants with 1× concentrations on surfaces (titanium, polyethylene, and stainless steel) against the *Staphylococcus aureus*. By comparing the results in all 2D images, there is a clear difference between the surface topography. The growth of *Staphylococcus epidermidis* was observed more compared to *Staphylococcus aureus*. The trend was further justified by 3D analysis, where coarser structures with higher topographic peaks were analyzed. The 3D images of Figure 9a–c show the thickness of *Staphylococcus epidermidis* growth (120 µm). It is the overall thickness of the material (AgNPs@BB extract coating + growth of *Staphylococcus epidermidis*). The same trend has been justified in a similar study, where a 3D examination of copper coating was analyzed [61].

The confocal analysis shows a green color at some points (growth of biofilm). However, the green color is less compared to uncoated implants. Moreover, there is less thickness of coatings (AgNPs@BB extract coating + growth of *Staphylococcus aureus*), which is 70 µm, 62 µm, and 50 µm. It means there is less growth of *Staphylococcus aureus* biofilm compared to *Staphylococcus epidermidis*. Though the coating concentration of 1× inhibits the growth of both bacteria, there remained biofilm (as shown by a partial green color).

Figure 10a,f shows the 2D and 3D confocal microscopic analysis of coated implants with AgNPs@BB extract concentrations of 10 mg/mlL. Figure 10a–c is the 2D and 3D confocal images of coated implants with 100 mg/mL concentrations (titanium, polyethylene, and stainless steel) against the biofilm formation of *Staphylococcus epidermidis ,* and Figure 10d–f is the 2D and 3D confocal images of coated implants with 2× concentrations on surfaces (titanium, polyethylene, and stainless steel) against the *Staphylococcus aureus*. The 3D images of Figure 10a–c shows the thickness of *Staphylococcus epidermidis* growth (80 µm, 80 µm, 50 µm). It is the overall thickness of the material (AgNPs@BB extract coating + zero growth of *Staphylococcus epidermidis*). The confocal analysis shows the black color at all points (which is zero growth of biofilm). It means 3D surface plots showed a significant reduction of the biofilms.

By comparing the results in all 2D images, there was a clear difference between the surface topography. The growth of *Staphylococcus epidermidis* was observed more compared to *Staphylococcus aureus*. The trend was further justified by 3D analysis, where coarser structures with higher topographic peaks were analyzed.

### 3.7. Biofilm Inhibition—Quantitative Assay

As stated earlier, there is a wider growth and spread of drug-resistant bacteria due to biofilm formation on orthopedic implants. This required identifying novel antibacterial methods to treat such bacterial infections and prevent biofilm formation. The current study aimed to evaluate the biofilm inhibitory activities of biogenic AgNPs@BB extract conjugate against *Staphylococcus aureus* and *Staphylococcus epidermidis* , which was carried out in 96 microtiter plate wells. The performance of biosynthesized AgNPs@BB extract composites in inhibition of biofilm formation was measured at the concentrations of 2, 4, and 8 µg/mL, equivalent to 0.5, 1, and 2 µg/mL against the standard *Staphylococcus aureus* and *Staphylococcus epidermidis* .

The OD (growth and biofilm density value) was found to be 0.25 (0.5 µg/mL), 0.05 (1 µg/mL), 0.02 (2 µg/mL) against Aureus, and 0.31 (0.5 µg/mL), 0.04 (1 µg/mL), 0.01 (2 µg/mL) for *Staphylococcus epidermidis* . Additionally, the negative control groups of both organisms showed a mean OD value of 0.01 for the case of AgNPs@BB extracts. Table 3 shows the concentrations of AgNPs@BB extract conjugates and inhibitions of the formation of biofilms of *Staphylococcus aureus* and *Staphylococcus epidermidis* when compared to the positive control (growth control). No biofilm formation inhibition was found for 0.5 µg/mL MIC concentration of AgNPs@BB extracts when compared to the (Ampicillin) positive control. The performance of AgNPs@BB extract conjugates with various concentrations toward the species of *Staphylococcus aureus and Staphylococcus epidermidis* in preventing the formation of biofilm by phenotypic assay is exhibited in Table 3.

The performance of AgNPs@BB extract conjugates with various concentrations toward the species of *Staphylococcus aureus and Staphylococcus epidermidis* in inhibition of the biofilm growth percentage by phenotypic assay is exhibited in Table 4. The results show the inhibition of biofilm growth at 94.2% ± 1.2SD and 96.2% ± 1.3SD against *Staphylococcus aureus and Staphylococcus epidermidis* , respectively.

### 3.8. Cytotoxicity Analysis

Figure 11a–e exhibits the results of the examination of the viability of cells (L929) grown on the surface of tested coatings for 24 h in varying concentrations of 5 µ/mL (control), 25µ/mL, 50 µ/mL, 75 µ/mL, and 100 µ/mL. Using the MTT assay to examine the viability, the obtained results indicated a high survival rate of fibroblast seeded on the coatings. The cellular viability levels were also seen to be higher by 5% compared to the control values. The aging and stability of AgNPs are important factors in their toxicity level. It has been reported that aged AgNPs incubated in water for six months increase in toxicity level, which is related to the silver ions released. Toxicity appears to be a cumulative effect of both silver ions and AgNPs, which is believed to be due to the released Ag ions, and others attribute toxicity to AgNPs [62].

However, no significant changes were recorded for the cells grown on the AgNPs@BB extracts sample in high concentration (75 µ/mL), as presented in Table 5. The cell viability (%) was 79.6 ± 2.7 with a * *p* <0.05. The AgNPs conjugate-treated L929 cells showed 68.488 ± 6.1% viability at the highest concentration (100 μg/mL) after 24 h of treatment. The viability of the cells of different samples tested well above the cut-off for cytotoxicity as recommended by ISO 10993-5:2009 (Biological Evaluation of Medical Devices Part 5) [63].

Table 5 indicates statistically significant differences in cell viability among the different concentration levels. Specifically, with the Tukey HSD post hoc test, intergroup statistically significant differences were observed (* *p* value ≤ 0.001). A statistically significant difference in cell viability was seen at a concentration of 5 µ/mL compared to concentrations of 25, 50, and 100 µ/mL, with corresponding *p*-values of 0.0413, 0.0088, and 0.0072, respectively. Likewise, statistically significant differences were also observed between other concentrations, i.e., 5 and 100 µ/mL (*p* = 0.0001), as well as between 25 and 100 µ/mL (*p* = 0.0127).

In this study, concentrations between 5 and 100 µg/mL of AgNPs@BB extracts supported the proliferation of L929 cell growth from hours using the MTT assay [64]. The assay is based upon the principle that MTT, a yellow water–soluble tetrazolium dye, is reduced by mitochondrial dehydrogenases to purple formazan crystals, which can be measured spectrophotometrically to calculate the number of viable cells left [65]. In order to move forward with the tools conventionally used to combat infections associated with orthopedic implants, in vitro studies must verify that each AgNP concentration exhibits antimicrobial activity lower than the cytotoxic concentration [66]. Usually, the cytotoxic species present in the implant coatings are equal to or lower than the cytotoxic concentration. In such a state, it can be safely concluded that the AgNPs@BB extract conjugate coatings are not the cause of cytotoxicity, especially when they encounter fibroblasts, but rather they might result in a consistent acceleration and regeneration of the bone in the case of orthopedic applications. In addition, the surface cytotoxicity of AgNPs showed how surface modulation can be used in the reduction of the cytotoxicity of AgNPs, which is consistent with the findings that the design and engineering of AgNPs could help mitigate any potential side effects [67]. 

## 4. Conclusions

In this study, AgNPs@BB extracts were developed in the form of thin film coatings for implantable surfaces of titanium (Ti), stainless steel (SS), and polyethylene (PE). Additionally, their chemical structure was confirmed by UV-spectrometry, DLS, and FTIR analyses. Surface micro topographies and mechanical properties were evaluated by SEM and TEM. When biofilm formation was assessed by confocal microscopy, the antimicrobial properties were also assessed in *Staphylococcus aureus* and *Staphylococcus epidermidis* cultures, as well as on their biofilm formation. The coatings of AgNPs@BB extract demonstrated a homogenous dispersion of elements across their surfaces. They exhibited moderate levels of roughness, as seen in microtopography with uncoated titanium (Ti) surfaces. All coatings displayed greater antimicrobial effects of AgNPs@BB extract (IX, 50 mg/mL), and concentration was associated with increased bacterial inhibition. Additionally, coatings loaded with AgNPs@BB extract (2×, 100 mg/mL) conjugates showed marked inhibition of planktonic bacteria and their biofilms, with near complete inhibition of *Staphylococcus aureus* and *Staphylococcus epidermidis* pathogens and of biofilm formation. These robust, retentive, and antimicrobial coatings of AgNPs@BB extracts for implantable titanium, stainless steel, and polyethylene surfaces of materials were based on dipping method processes, which may potentially afford a novel strategy for biomaterial applications.

These findings equate to a sort of microbiological characterization to confirm the presence of a potent antibacterial activity of AgNPs@BB extract conjugates, demonstrating ideal cytocompatibility properties. However, owing to certain limitations of this study, this observation needs further investigations of the role of AgNPs@BB extract and alternatives when applying it to orthopedics implant biomaterials.

## Figures and Tables

**Figure 1 antibiotics-12-01403-f001:**
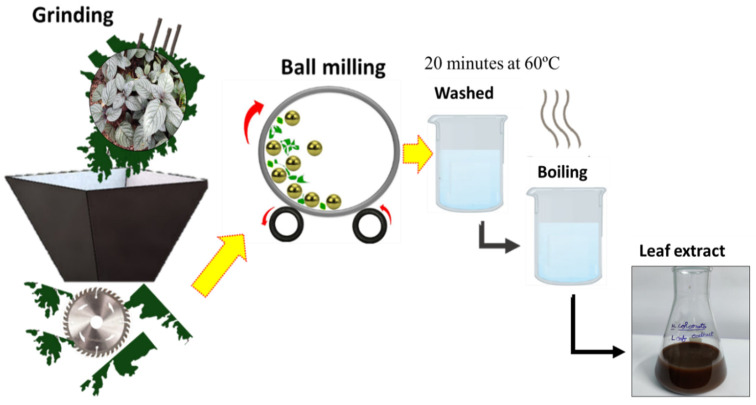
Flow process for the preparation of the *Hemigraphis colorata* leaf extract.

**Figure 2 antibiotics-12-01403-f002:**
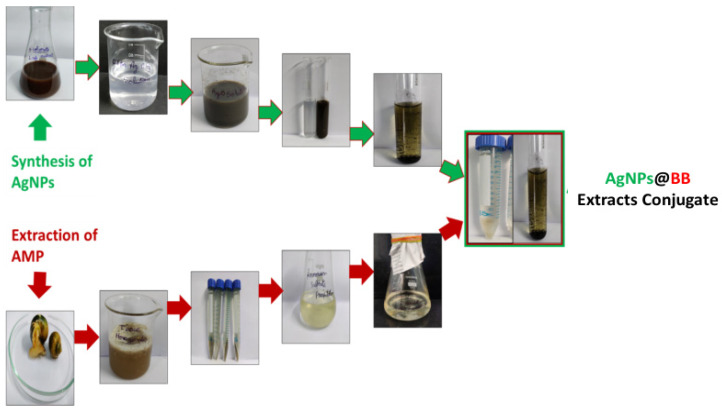
Schematic preparation of AgNPs, BB extract, and AgNPs@BB extract composites. Steps showing parallel action of AMP extraction and synthesis of AgNPs, ending up with conjugation of AgNPs@BB extract conjugates.

**Figure 3 antibiotics-12-01403-f003:**
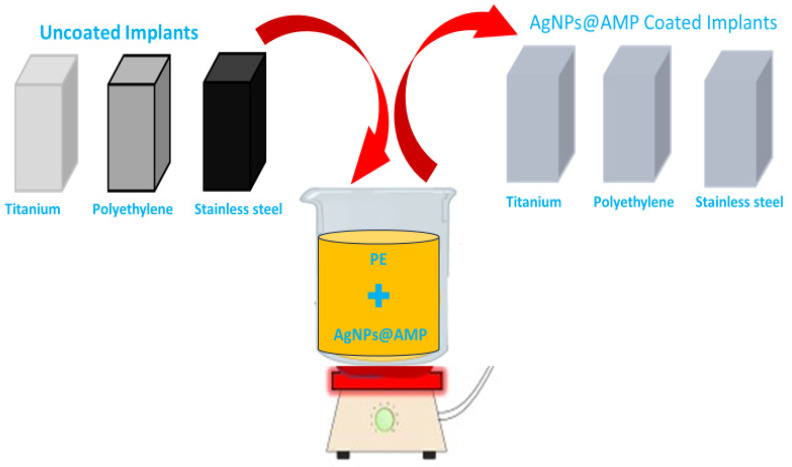
Scheme for the coating of AgNPs@BB extract composites on implants.

**Figure 4 antibiotics-12-01403-f004:**
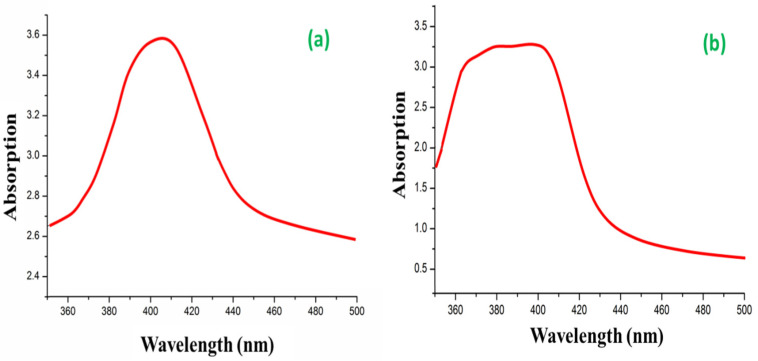
UV visible spectroscopy of (**a**) AgNPs and (**b**) AgNPs@BB extract.

**Figure 5 antibiotics-12-01403-f005:**
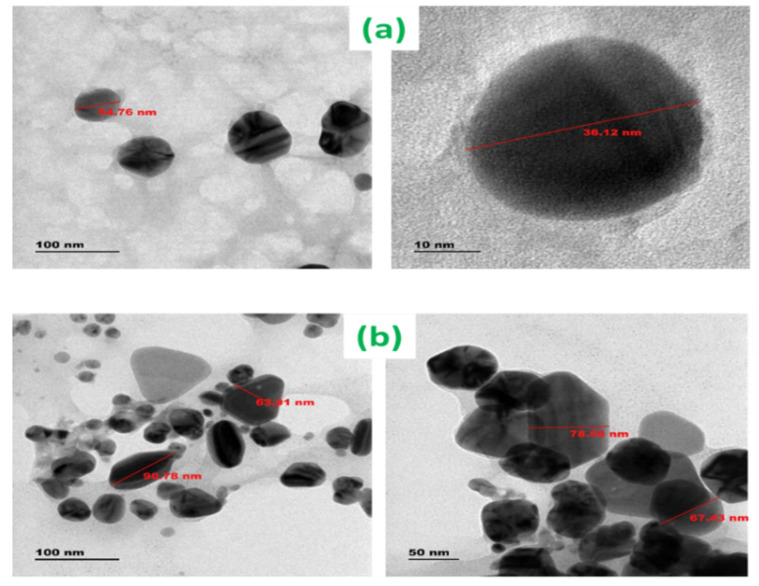
TEM analysis of (**a**) AgNPs, (**b**) AgNPs@BB conjugates, absorption spectrum in the range of 350–800 nm.

**Figure 6 antibiotics-12-01403-f006:**
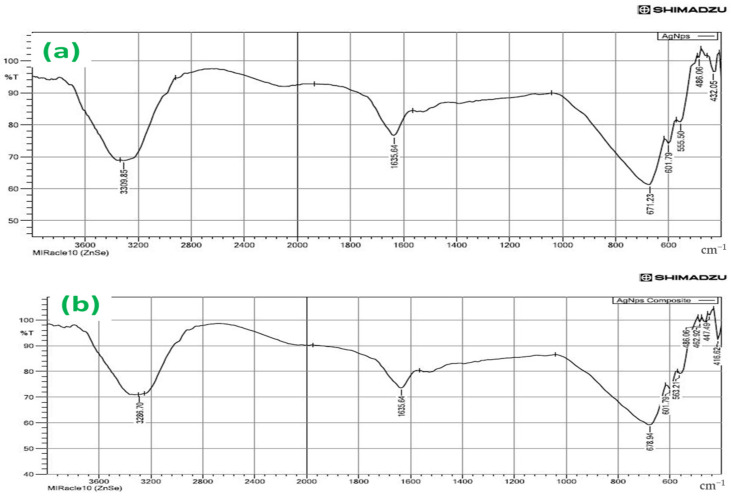
FTIR spectra of (**a**) AgNPs and (**b**) AgNPs@BB extract conjugates.

**Figure 7 antibiotics-12-01403-f007:**
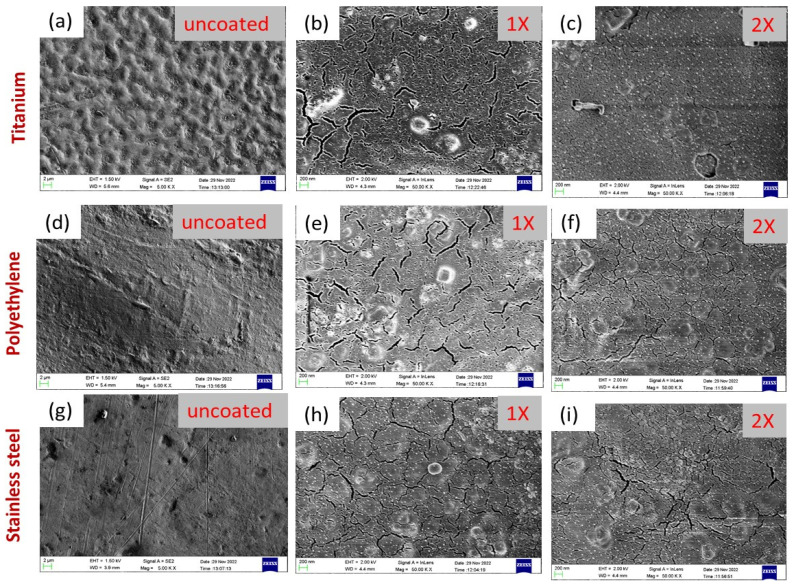
SEM analysis of (**a**) uncoated titanium, (**b**,**c**) titanium coated with 1× and 2× concentration of AgNPs@BB extract, (**d**) uncoated polyethylene, (**e**,**f**) polyethylene coated with 1× and 2× concentration of AgNPs@BB extract, (**g**) uncoated stainless steel, and (**h**,**i**) stainless steel coated with 1× and 2× concentration of AgNPs@BB extract conjugates.

**Figure 8 antibiotics-12-01403-f008:**
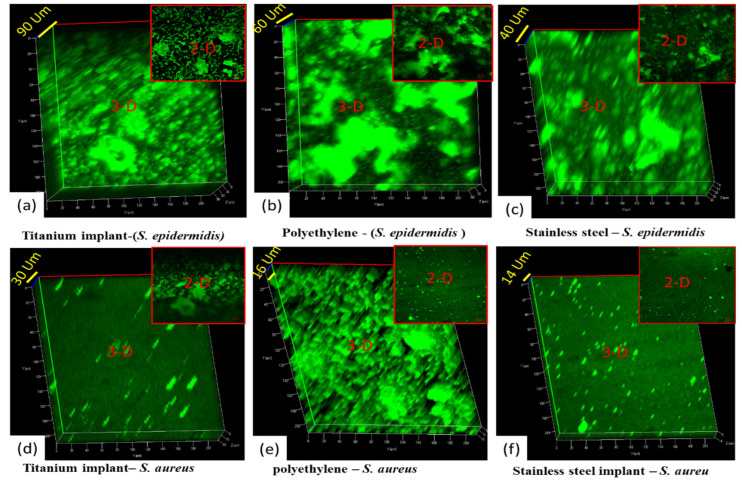
**Two-dimensional and three-dimensional** images of biofilm formation of *Staphylococcus epidermidis* on uncoated implants of (**a**) titanium, (**b**) polyethylene, and (**c**) stainless steel, and 2D and 3D images of biofilm formation of *Staphylococcus aureus* on uncoated implants of (**d**) titanium, (**e**) polyethylene, and (**f**) stainless steel.

**Figure 9 antibiotics-12-01403-f009:**
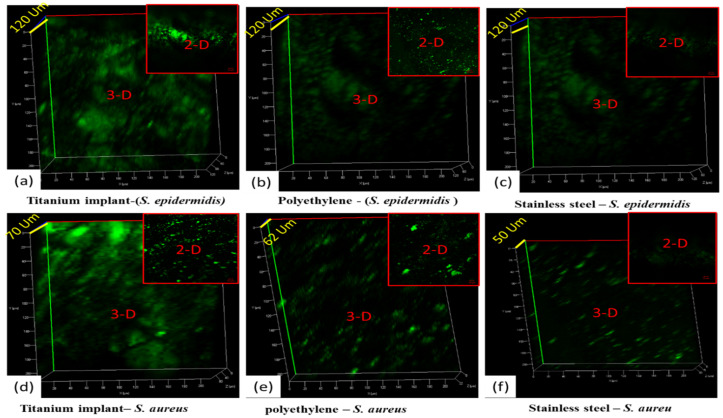
**Two-dimensional and three-dimensional** images of biofilm formation of *Staphylococcus epidermidis* on 1× concentration (AgNPs@BB extract) coated implants of (**a**) titanium, (**b**) polyethylene, and (**c**) stainless steel, and 2D and 3D images of biofilm formation of *Staphylococcus aureus* on 1× concentration (AgNPs@BB extract) coated implants of (**d**) titanium, (**e**) polyethylene, and (**f**) stainless steel.

**Figure 10 antibiotics-12-01403-f010:**
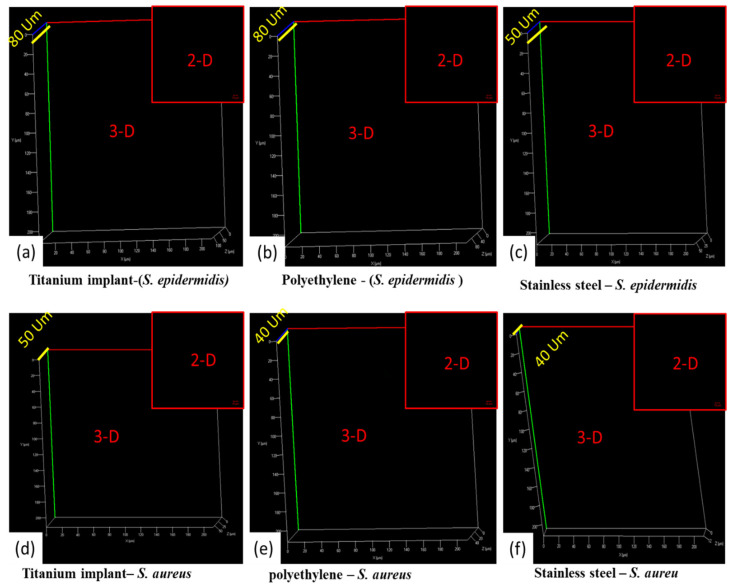
**Two-dimensional and three-dimensional images** of biofilm formation of *S. Epidermidis* on 2× concentration (AgNPs@BB extract) coated implants of (**a**) titanium, (**b**) polyethylene, and (**c**) stainless steel, and 2D and 3D images of biofilm formation of *Staphylococcus aureus* on 2× concentration (AgNPs@BB extract) coated implants of (**d**) titanium, (**e**) polyethylene, and (**f**) stainless steel.

**Figure 11 antibiotics-12-01403-f011:**
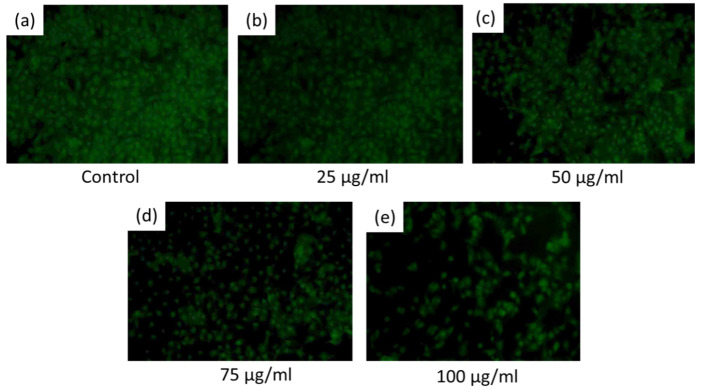
Acridine orange staining for examining cytotoxicity of the AgNPs@BB extract conjugates.

**Table 1 antibiotics-12-01403-t001:** Major FTIR peaks and associated functional groups.

Peaks	Functional Groups
3309 cm^−1^	Hydroxyl group (-OH) stretching vibrations
3100 cm^−1^	Unsaturated C-H stretching vibration of AMP
1635 cm^−1^	Carbonyl (C=O)
1200–1300 cm^−1^	Aromatic -CH bending vibrations
1000–1100 cm^−1^	Phenolic (C-O-H stretching vibrations)

**Table 2 antibiotics-12-01403-t002:** Resazurin-based MIC assay of AgNPs, BB extract, and AgNPs@BB extract conjugate against *Staphylococcus aureus and Staphylococcus epidermidis*.

MIC (μg/mL)		
	*Staphylococcus aureus*	*Staphylococcus epidermidis*
AgNPs	7.8 ± 0.67	12.8 ± 0.67
BB extract	31.2 ± 0.01	15.6 ± 4.3
AgNPs@BB extract conjugate	3.9 ± 0.04	3.9 ± 0.01

**Table 3 antibiotics-12-01403-t003:** Biofilm growth (OD Value) on treatment with different concentrations of AgNP@BB conjugates.

Pathogens	Concentrations (µg/mL)	OD Values			Mean
		1	2	3	
*Staphylococcus aureus*	Negative control	0.01	0.01	0.02	0.01
	Positive control	0.38	0.43	0.58	0.46
	0.5	0.25	0.21	0.26	0.24
	1	0.05	0.06	0.05	0.05
	2	0.02	0.02	0.03	0.02
*Staphylococcus epidermidis*	Negative control	0.01	0.01	0.01	0.01
	Positive control	0.53	0.42	0.38	0.44
	0.5	0.31	0.32	0.27	0.3
	1	0.04	0.03	0.04	0.03
	2	0.01	0.02	0.02	0.01

**Table 4 antibiotics-12-01403-t004:** Inhibition of biofilm growth by AgNPs@BB extract conjugates.

Pathogens	Concentrations (µg/mL)	Biofilm Inhibition (%) *
*Staphylococcus aureus*	0.5	47.4 ± 5.7
	1	87.7 ± 1.2
	2	94.2 ± 1.2
*Staphylococcus epidermidis*	0.5	32.3 ± 5.9
	1	91.7 ± 1.3
	2	96.2 ± 1.3

* Note: Values are represented in MEAN ±SD.

**Table 5 antibiotics-12-01403-t005:** Cell viability after treatment with AgNPs@BB extract conjugates (%).

Concentration (µg/mL)	Cell Viability (%)
5	96.3 ± 3.2
25	83.8 ± 5.6
50	80.1 ± 3.8
75	79.6 ± 2.7
100	68.5 ± 6.1

Note: One-way ANOVA (*F* = 14.8736, *df* = 4, CI = 95%).

## Data Availability

The authors confirm that the data supporting the findings of this study are available within the article.

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
