# Peer review of "Deploying a Novel Approach to Prepare Silver Nanoparticle Bellamya bengalensis Extract Conjugate Coating on Orthopedic Implant Biomaterial Discs to Prevent Potential Biofilm Formation"

_antibiotics, 2023, doi:10.3390/antibiotics12091403_

Round 1

Reviewer 1 Report (New Reviewer)

The article of Shafqat Qamer et al is very interesting and voluminous research, which described the obtaining and characterization of AgNPs conjugated with antimicrobial peptides, and the study of their properties as coatings for some surfaces (which act as orthopedic implants) against microbial films fouling. The work was performed with use of wide range of research methods. The study of antibacterial and antifouling properties of AgNPs is perfectly described. The excellent result of antimicrobial activity of conjugated AgNPs on the different surfaces was obtained. However there are some annoying disadvantages.

1)      The name of the article is not correct. In my mind, the word “anti” is superfluous

2)      Authors should add some references after the phrase «To the best of our knowledge there are only few studies available on surface coating of implants with AgNPs@BB extract conjugates».

3)      Authors should quote reference 65 in order, that is, assign the number 41 in the phrase

“For the purpose of cell cytotoxicity of AgNPs@BB extracts, mouse fibroblastic cells (L929) were cultured, following the supplier guidelines. Cytotoxicity of the synthesized AgNPs@BB composite as assessed by MTT (3- [4,5-dimethylthiazole-2-yl]-2, 5-diphenyl tetrazolium bromide) dye conversion assay [40, 65]”.

4)      Authors should check the correspondence of references to their mention in the text, apparently already in the Introduction section there was confusion

5)      In section 3.2 ТЕМ images of AgNPs@BB conjugates (Figure 5b) demonstrate the presence of triangular NPs. Usually in the case of such NPs their SPR maximum is red shifted due to the larger effective dipole moment than of spherical NPs (see, for example, doi:10.1016/j.molliq.2011.10.002), but it is not reflect on Figure 4b. Please, add the view of absorption spectrum in range of 350-800 nm and/or specify the reason of the absence of a band or shoulder in the region of 600 nm.

6)      In section 3.3 the displacement of band at 1635.64 cm−1 during the formation of AgNPs@BB conjugates is described as an evidence of interaction of AgNPs and antimicrobial peptides, however there is no displacement on Figure 6. Please, describe the IR spectra more clear.

7)      The Figure 7, on which Authors link in section 3.4, is absent, and after Figure 6 the next is Figure 9. Please, correct.

Author Response

dear Reviewer 1

Please find attached correction table prepared as per your kind comments

Reviewer 2 Report (New Reviewer)

In this manuscript, the authors developed the AgNPs@BB extract conjugate to endow implantable devices antibacterial and inhibition of biofilm abilities. Interestingly, the authors extracted antimicrobial peptides from antimicrobial peptides from B. bengalensis and prepared the AgNPs via a biosynthesis process using silver nitrate and leaf extraction. Then the authors prepared the AgNPs@BB extract conjugate with the above ingredients. After coating the AgNPs@BB extract conjugate on the surface of different substrates (titanium, polyethylene, or stainless steel), the coated substrates showed good antimicrobial properties and biofilm controlling ability. Though the contents are interesting, the authors need to address my comments before publishing this manuscript.

Comments:

Comment 1: In the introduction section, paragraphs one and two are relatively too long. Refine these two paragraphs so that readers can better focus on the biofilm contents starting from the 3rd paragraph.

Comment 2: In introduction paragraph 4, add some example references to support the point: “Studies have recommended antimicrobial strategies based on one of the two methods:”.

Comment 3: Overall, the introduction can be more concise by removing some not relevant contents.

Comment 4: Add more information in the caption of figure 2 so that readers can understand the listed steps without looking into the text.

Comment 5: In figure 3, the uncoated objects are the same color though they are made from different sources (titanium, polymer, or stainless steel). After coating (I assume the coating reagent is the same), why the indication colors of these three objects are different? Do they have different surface properties? In my opinion, the 3 objects original color should be different as they are different materials, and the coated objects should have the same color as they have the same surface properties. Is my speculation correct?   

Comment 6: Figure 11 and figure 13 images look blurry. The authors need to double check their confocal/fluorescent microscope techniques to make sure high-quality images were obtained. Plenty of confocal cell images can be found online and most of them show clear cell structure. I understand that this study is investigating biofilms which can be considered as colonies of bacteria. But still, confocal images are capable of capturing the 3-d structure of the biofilm. The blurry images in figures 11 and 13 are not satisfying.

Comment 7: There are many spelling or grammar errors. The authors should go over the manuscript many times to correct them. It is recommended that asking a native speaker to check the language after revisions.

There are many grammer or spelling errors. The help from a native speaker should be helpful to improve the language quality.

Author Response

Dear Reviewer 2

Please find attached correction table prepared as per your kind comments

Thanks

Round 2

Reviewer 2 Report (New Reviewer)

The authors addressed my concerns. My only suggestion is to go over this manuscript to correct any possible spelling errors. For example, the AgNO3, where 3 should be subscript. 

This manuscript is a resubmission of an earlier submission. The following is a list of the peer review reports and author responses from that submission.

Round 1

Reviewer 1 Report

The manuscript entitled "Deploying A Novel Approach to Prepare the AgNPs@AMP Conjugate Coating On Different Orthopaedic Implant Biomaterial Discs for Potential Anti Biofilm Formation" deal with the development of nanocomposite coatings for orthopedic implants. Further, a detailed analysis of the antibacterial efficiency was carried out.  The obtained results suggested the potential use of as-developed coatings. However, a major drawback of this study is the lack of statistical analysis. The addition of statistical analysis results will greatly enrich the quality of this work. I recommend authors carry out non-parametric tests along with analyses of variance.

Author Response

Dear reviewer,

Thanks  you for your useful comments that helped me to improve the quality of the article. all comments have been addressed and the article have been modified. the changes are highlighted in the modified article the reviewer response sheet have been attached below and the full article have been mailed to the editor.

Shafqat Qamer( first author)

Reviewer 2 Report

The Authors investigated the efficacy of the antibacterial and antibiofilm activities of AMP- conjugated AgNPs extracted from Bellamya bengalensis on 96-wells microplates and biomaterial discs.

The chosen topic is of scientific interest and is corresponding to the scope of the Journal Antibiotics.

The abstract should be abbreviated and contain a maximum of 200 words instead of 377. The discussion and results must be written separately. Please, follow the Authors’ instructions.

The manuscript contains several serious improprieties that Authors should deal and solve.

The Authors refer about AMPs in their work, but I would argue that they used a pool of extracted proteins than specific antimicrobial peptides. The Authors have not identified the bioactive constituents performing techniques based on the isolation and characterization of the peptides. For example, Gauri S.S. et al. (2011) discovered an antimicrobial peptide of 1676 Da from Bellamya bengalensis using ultrafiltration and reversed phase liquid chromatography.

I have concerns about experimental protocols. Some parts are not well written/exposed, and some parts are not accurate. Please, consider the following remarks.

To study the AgNPs@AMP susceptibility, the Authors used two bacterial strains belonging to the same genus Staphylococcus. Why did not they consider using other microorganisms involved in implant related infections such as Enterococcus spp. or one Gram-negative representative?

The Authors performed technical triplicate, but they should have performed at least three biological independent tests (3 plates for each type in different times) which are significant above all for the obtained result. For MIC analysis, the Authors obtained different values for the same antimicrobial concentration in the AgNPs, AMP and AgNPs@AMP microplate.

Another observation is why did not the Authors perform the MBC (minimum bactericidal concentration) to make their results more comprehensible and complete.

The Authors referred the microtiter assay method to the CLSI M100-S22 document (supplemental tables for MIC testing) but the protocol information to follow for the microdilution method belong to the M07 and M11 documents. I advise the Authors be more precise when they expose the protocol referring to the antimicrobial activity.

Furthermore, I did not understand how the Authors provided the results of the MIC. Can they add a reference? In my opinion MIC cannot match the concentration where 1 or 2 out of 3 wells have bacterial growth. The graphic where the O.D. results are plotted is missing and Figure 7 for a merely qualitative outcome is not adequate.

There are some minor inaccuracies, i.e., “SYTO 99” (maybe is SYTO 9?), “0,5% McFarland” (are the Authors confident?), resazurin solution (which kit or reference did the Authors use?), “following the guidelines and industry trends” (what do the Authors mean?).

Figure 7. Please check the positive and negative control.

2.8 Cell cytotoxicity assay: The Authors incorrectly referred to the Figure 12. Moreover, there is no figure or table regarding the cell cytotoxicity assay result.

The English style needs to be revised to make the manuscript more formal. Some sentences must be rephrased.

Author Response

Dear Reviewer 2

Thank you for your useful comments that helped me to improve the quality of the article. All comments have been addressed and the article has been modified. The changes are highlighted in the modified article. The reviewer response sheet has been attached below and the full article has been mailed to The Editor.

Shafqat Qamer (First Author)

Reviewer 3 Report

This study investigates the effectiveness of the the AgNPs@AMP Conjugate Coating On Different Orthopaedic Implant Biomaterial Discs for Potential Anti Biofilm Formation design of a biodegradable locking compression However, the manuscript in the present formrequires some modifications. Some of the lacunae and possible modifications are given below.

1.     In the introduction section, the authors have briefly explained the literature on using Prosthetic Joint Infection (PJI) implants. However, keeping the objective of the current study in mind, detailed literature about the various n antibacterial AgNPs@AMPs coating Prosthetic Joint Infection (PJI) implants is missing. Authors are encouraged to add a specific discussion on it.

2.     Authors have assumed the antibacterial properties of the coated and uncoated implants but did not specify the durability of the coatings in terms of adhesion tests and comments on the biocompatibility behaviour of the coatings.

3.     In section 2.4, under the results and discussion section, it is mentioned the reasoning behind the antibacterial mechanism; it would be better if authors could provide the schematic of the mechanism

There are some grammatical errors which need to be taken care 

Author Response

Dear Reviewer 3

Thank you for your useful comments that helped me to improve the quality of the article. All comments have been addressed and the article has been modified. the reviewer response sheet has been attached below and the full article has been mailed to the editor.

Shafqat Qamer (First Author)

Round 2

Reviewer 2 Report

Point 2: The Authors correctly amended the Abstract.

Point 3: My concern is that the Authors named AMP in their study. It should be better to expose to the readers that they work with a proteins’ mixture and not with a specific antimicrobial peptide. In general, the other authors refer to AMP with antimicrobial peptide/peptides and not a proteins’ mixture. Then if the Authors cannot use specific AMP cause of expensive process, they should be careful and avoid using AMP term. Please rephrase the critical words and concept in the manuscript.

Point 4: The Authors did not improve the experimental section. 1. I understand, and I accept the reason why the Authors used only Staphylococcus species. 2. The Authors did not complete the 3 independent tests, but they only performed 3 wells in the same experimental test. The independent tests are useful especially when we obtain different results.

Point 5: I do not agree with the Authors answer. MBC is the minimum bactericidal concentration that demonstrates the lowest level of antimicrobial agent resulting in microbial death while MIC is the minimum inhibitory concentration that demonstrates the lowest level of antimicrobial agent that inhibits growth. The two results can help you to have a complete view especially when you obtain different results.

Point 6: Can the Authors be more precise when they write the protocol of the MIC test? For example, the Authors reported that they used “the concentration of 106 CFU/ml” but after they reported “2 g positive S. aureus and S. epidermidis”. 2 g is not clear. Which in vitro toxicology assay kit did they use?

Point 7: I do not yet understand how the Authors gave the MIC result.

Just as example:

Figure 7, in F line AgNPs@AMP susceptibility showed that for Staphylococcus aureus there was growth for the 7.8 ug/ml (column 7), no growth for 3.9 ug/ml (column 8), and growth again for 1.9 ug/ml (column 9)  (this is an example of unexpected data, and this is the reason why I asked the triplicates); in the G line there was no growth in column 7 and growth from 3.9 ug/ml (column 8) and beyond; in the H line there was no growth in column 8 and growth from 1.9 ug/ml (column 9) and beyond. The Authors’ decision was to provide the 3.9 ug/ml value as the MIC. Can the Authors explain in the Materials and Methods the reason of the MIC choice?

Author Response

Open Review Response:

  1. Introduction has been “sufficiently” provided information and references updated.
  2. References thoroughly checked and all references were found relevant
  3. Research design has been cross-checked, triangulated, and verified form other studies. Hence, there is no scope of any amendment if changed, the paper’s robustness will be disturbed)
  4. According to all authors, the methods are adequate and parallel to similar studies. Anything extra would disturb its integrity and credibility
  5. Same as methods, the results are found to be accurate, compact and focused; however, a few changes have been made based on the comments given in Point 6 and 7 below.
  6. According to authors, all conclusions drawn are consistent with the previous results.

 Comments and Suggestions for Authors

  • Point 2: The Authors correctly amended the Abstract.

Response:  Thanks for accepting the amendment.

  • Point 3: My concern is that the Authors named AMP in their study. It should be better to expose to the readers that they work with a proteins’ mixture and not with a specific antimicrobial peptide. In general, the other authors refer to AMP with antimicrobial peptide/peptides and not a proteins’ mixture. Then if the Authors cannot use specific AMP cause of expensive process, they should be careful and avoid using AMP term. Please rephrase the critical words and concept in the manuscript.

Response: 

We fully agree with the reviewer’s comment that our accountability is towards the readers to provide them a clear picture of our work. In this connection, we wish to submit: firstly, this term is rather used as a common and generic term; secondly, AMPs are of course protein mixtures, but no specific AMP was mentioned in my whole manuscript (as in other studies e.g., Defensins and Cathelicidins which have broad-spectrum antimicrobial activities against bacterial pathogens); thirdly and most importantly, AMPs differ from protein mixtures as they have antibacterial activity, which no protein mixtures might have. By changing critical words or removing AMPs from the article would change the whole meaning and purpose. 

Thank you first for this observation and considering it an important factor. I hope I am able to convince you with my comments

  • Point 4: The Authors did not improve the experimental section. I understand, and I accept the reason why the Authors used only Staphylococcus species. 2. The Authors did not complete the 3 independent tests, but they only performed 3 wells in the same experimental test. The independent tests are useful especially when we obtain different results.

Response:

We understood the reviewer’s argument and respect the same too. However, we may humbly state that: first, we performed tests on three different materials (Ag, AMP, AgNPs@AMP) in three different experimental wells. They were not the same experimental well, but with same concentration. The objective was to check which MIC is better in same concentration. Secondly, it may also be emphasized that resazurin MIC method testing was repeated in triplicate (no independent test was performed nor required), still results were same and validated. (The same has been rephrased for clarity in the manuscript (lines 313 to 325)

  • Point 5: I do not agree with the Authors answer. MBC is the minimum bactericidal concentration that demonstrates the lowest level of antimicrobial agent resulting in microbial death while MIC is the minimum inhibitory concentration that demonstrates the lowest level of antimicrobial agent that inhibits growth. The two results can help you to have a complete view especially when you obtain different results.

Response:

We fully support the reviewer’s view about MBC and MIC. The reviewer has rightly stated the purpose of the two. However, as stated in the previous comment (Point 4) the three tests conducted were not in the same experimental well, though with same concentration, where the purpose was to check which MIC is better in the same concentration. It was not our objective to compare the two results (e.g., MBC and MIC). This point is further explained later in point 7 where a retest was conducted and the discrepancy was corrected. (The same has been rephrased for clarity in the manuscript (lines 313 to 325)

  • Point 6: Can the Authors be more precise when they write the protocol of the MIC test? For example, the Authors reported that they used “the concentration of 106CFU/ml” but after they reported “2 g positive  aureus and S. epidermidis”. 2 g is not clear. Which in vitro toxicology assay kit did they use?

Response:  I corrected all typo errors and accepted the reviewers’ comment

  • Point 7: I do not yet understand how the Authors gave the MIC result.

Just as example:

 Figure 7, in F line AgNPs@AMP susceptibility showed that for Staphylococcus aureus there was growth for the 7.8 ug/ml (column 7), no growth for 3.9 ug/ml (column 8), and growth again for 1.9 ug/ml (column 9)  (this is an example of unexpected data, and this is the reason why I asked the triplicates); in the G line there was no growth in column 7 and growth from 3.9 ug/ml (column 8) and beyond; in the H line there was no growth in column 8 and growth from 1.9 ug/ml (column 9) and beyond. The Authors’ decision was to provide the 3.9 ug/ml value as the MIC. Can the Authors explain in the Materials and Methods the reason of the MIC choice?

Response: 

Thank you for pointing out this discrepancy and it has been corrected (line No 561 to 565 and Figure 7 (a,b,c), replaced after re-test).  

The F line was earlier showing growth in column 7 but no growth in column 8 but again growth in column 9. To rectify this, the test was repeated and the desired results were obtained in column 7 and 8. The results are same in column 7 and 8 got in triplicate which were 7.8 ug/ml and 3.9 ug/ml respectively.  However, in column 9, as correctly pointed out by the reviewer, there was no growth because after column 8, double dilution resulted in the decrease of MIC. The column 9 showed growth because this concentration (1.9 ug/ml) cannot inhibit the growth.

 Comments on the Quality of English Language: None

Reviewer 3 Report

The manuscript can be accepted in the present form 

Author Response

The reviewer has accepted the corrections in Round 1 and recommended the publication of this paper